# A Network Analysis on Cloud Gaming: Stadia, GeForce Now and PSNow

**Andrea Di Domenico** [1]**, Gianluca Perna** [2,*]**, Martino Trevisan** [2]**, Luca Vassio** [2] **and Danilo Giordano** [1,*]

1 Department of Control and Computer Engineering, Politecnico di Torino, Corso Duca degli Abruzzi, 24, 10129 Torino, Italy; andrea.didomenico@studenti.polito.it
2 Department of Electronics and Telecommunications, Politecnico di Torino, Corso Duca degli Abruzzi, 24, 10129 Torino, Italy; martino.trevisan@polito.it (M.T.), luca.vassio@polito.it (L.V.)
* Correspondence: gianluca.perna@polito.it (G.P.); danilo.giordano@polito.it (D.G.)

**Abstract:** Cloud gaming is a class of services that promises to revolutionize the videogame market. It allows the user to play a videogame with essential equipment while using a remote server for the actual execution. The multimedia content is streamed through the network from the server to the user. Hence, this service requires low latency and a large bandwidth to work properly with low response time and high-definition video. Three of the leading tech companies (Google, Sony, and NVIDIA) entered this market with their products, and others, like Microsoft and Amazon, are also launching their platforms. However, these companies have released little information about their cloud gaming operation and how they utilize the network. In this work, we study cloud gaming services from the network point of view. We collect more than 200 packet traces under different application settings and network conditions from a broadband network to poor mobile network conditions, for 3 cloud gaming services, namely Stadia from Google, GeForce Now from NVIDIA and PS Now from Sony. We analyze the employed protocols and the workload that they impose on the network. We find that GeForce Now and Stadia use the RTP protocol to stream the multimedia content, with the latter relying on the standard WebRTC APIs. Depending on the network and video quality, they result in bandwidth-hungry services consuming up to 45 Mbit/s. PS Now instead uses only undocumented protocols and never exceeds 13 Mbit/s. 4G mobile networks can often sustain these loads, while traditional 3G connections struggle. The systems quickly react to deteriorated network conditions, and packet losses up to 5% do not cause a reduction in resolution.

**Keywords:** cloud gaming; network measurements; mobile networks

## 1. Introduction

Year by year, new services are born as the Internet connection becomes faster and more reliable for end users [1]. The Hypertext Transfer Protocol (HTTP) has been the foundation of the World Wide Web, allowing the transfer of static content through the network. In the late 1990s and early 2000s, other protocols have become popular while supporting a broader spectrum of services. Notable examples are peer to peer (P2P) for sharing files and voice over IP (VoIP) for phone calls, made popular by Skype. In the 2010s, social networks and high-definition video streaming have become the new heavy-hitters in terms of involved users and traffic volume, making the Internet a pillar of our working and leisure activities. Such heterogeneous classes of services have different requirements from the network point of view. For example, VoIP requires low latency, while video streaming requires a large bandwidth. Their popularity shows that the Internet has succeeded in providing a reliable means for these applications to run, and satisfying the users' expectations.

During the last couple of years, some of the largest tech companies like Google and Sony announced new platforms for so-called cloud gaming. In a few words, the user plays a videogame using their equipment, while the actual execution takes place on the cloud, and the multimedia content is streamed through the network from the server to the

user. This paradigm implies that the user terminal no longer needs to be equipped with powerful hardware, like graphic processing units (GPUs), to play with recent demanding games, but only requires a fast and reliable communication with the cloud. These new platforms promise to revolutionize the videogame industry, which has steadily increased in recent decades from basically a non-existing market in the 1970s to an estimated USD 159.3 billion sales in 2020, and 2.7 billion players worldwide [2]. The videogame sector is even estimated to be the most lucrative sector in the whole entertainment market, recently surpassing television. Given these facts, the success of cloud gaming services would be of massive importance for the companies promoting it and for the Internet network handling the resulting traffic.

Google, NVIDIA, and Sony announced three different proprietary platforms for cloud gaming, namely, Stadia, GeForce Now, and PS Now. When writing this paper (September 2021), these services are available to users as a complete product. Microsoft also proposed its platform named xCloud, which is currently being tested. Amazon recently announced Luna, which is currently under testing as well. The companies disclosed a limited amount of information about the infrastructure of such services and how they utilize the network in terms of employed protocols and bandwidth usage.

In this work, we target the three cloud mentioned gaming services, and offer a preliminary characterization of the network workload that they generate. We collect 225 packet traces under different application settings and network conditions. Leveraging these data, we show which protocols that each application adopts for streaming multimedia content, the volume of traffic that they generate, and the servers and domains that they contact (for brevity, we use, throughout the paper, the term domain referring to the fully qualified domain name). We show that Stadia and GeForce Now adopt the standard real-time protocol (RTP) for streaming [3], largely adopted by real-time communication applications [4], while PS Now uses an undocumented protocol or encapsulation mechanism. Stadia and GeForce Now transmit video up to 45 Mbit/s, while PS Now does not exceed 13 Mbit/s. For comparison, a Netflix 4K video consumes approximately 15 Mbit/s (https://help.netflix.com/en/node/87, accessed on 1 October 2021), making cloud gaming potentially a new heavy hitter of the future Internet. With these characteristics, mobile networks can sustain high-definition gaming sessions only in the case of good-quality 4$G$ connections, as we quantify using real-world speedtest measurements. Their client applications contact cloud servers located in the autonomous systems (ASes) of the respective corporations. They are no further away than 20 ms in terms of round trip time from our testing location in northern Italy, a figure low enough for playing a videogame with no noticeable lag. To the best of our knowledge, we are the first to perform a study on these cloud gaming services, namely Stadia, GeForce Now, and PS Now, showing their characteristics and peculiarities in terms of network usage.

The remainder of the paper is organized as follows. Section 2 presents the related work. Section 3 describes our experimental setup, while Section 4 illustrates the findings that we obtain while analyzing the packet traces. Finally, Section 5 concludes the paper. To let other researchers replicate and extend our results, we release sample packet traces available at [5].

## 2. Related Work

Several research papers have already studied cloud gaming under different perspectives [6]. Authors in [6] focused on the study of cloud gaming platforms and optimization techniques. At the same time, several works provided general frameworks and guidelines for deploying cloud gaming services from the technical [7,8] and business [9] points of view. Other studies focused on the factors affecting the subjective QoE of users [10,11] or proposed novel techniques to improve it [12,13]. Some works proposed approaches for optimizing multimedia streaming [14–16] or GPU usage [17] in the specific context of cloud gaming. Interestingly, in 2012, the authors of [18] concluded that the network infrastructure

was unable to meet the strict latency requirements necessary for acceptable gameplay for many end-users.

Industrial pioneers of cloud gaming such as Onlive, StreamMyGame, and Gaikai have already been the object of study [19–22]. However, research papers targeting the services recently launched by leading tech companies such as Google and Sony have not yet been published. Only Stadia has been studied by Carrascosa et al. [23], who investigated its network utilization; however, their work did not compare with other platforms, nor explore mobile scenarios. This paper closes this gap, and first characterises the generated traffic in terms of employed protocols and network workload.

## 3. Measurement Collection

In this section, we describe our experimental testbed and the dataset that we collect. We focus on three cloud gaming services, namely Stadia, GeForce Now, and PS Now, on which we created standard accounts that we use for our experiments. We deploy a testbed using three gaming devices: a PC running Windows with a 4K screen, an Android smartphone, and the dedicated Stadia dongle.

All devices are connected to the Internet through a Linux gateway equipped with two 1 Gbit/s Ethernet network interfaces and a 300 Mbit/s WiFi 802.11ac wireless card. The Windows PC is connected to the gateway on the first Ethernet card, while the Android smartphone and the Stadia dongle are connected via WiFi. The gateway uses the second Ethernet interface as an upstream link to the Internet, provided by a 1 Gbit/s Fiber-To-The-Home subscription located in Turin, Italy. Figure 1 sketches our testbed. Stadia runs via the Chrome browser on the Windows PC and its mobile application on the Android phone (we used version 2.13). Moreover, we perform additional experiments using the dedicated Chromecast Ultra dongle, which allows the user to play and connect it to a screen. GeForce Now runs from a specific application in both cases (version 1.0.9 for PC and 5.27.28247374 for Android), while PS Now only works from the PC application (version 11.0.2 was used).

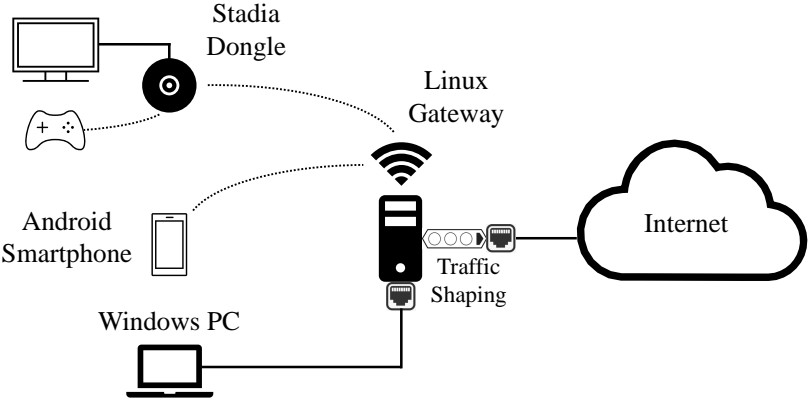

**Figure 1.** Testbed used for the experimental campaigns.

We play the three services making gaming sessions approximately 5–10 min long and capturing all the network traffic that the devices exchange with the Internet. We seek reliable results by playing a broad spectrum of video games on all platforms, from first-person shooters to racing and adventure—e.g., Sniper Elite 4, Destiny, Grid, and Tomb Raider.

With this testbed, we perform five different experimental campaigns, summarized in Table 1. Firstly, we run different gaming sessions for each platform using the Windows PC, the smartphone and the dongle (when possible). Secondly, we run different gaming sessions by using the Windows PC, and by manually configuring the applications to stream video with different quality levels. This option is available on Stadia and GeForce Now. Thirdly, only for GeForce Now, do we use the Windows application to use one of the 14 available

data centres by tuning the server location application setting. Next, we artificially impair the network in terms of bandwidth and latency and packet loss to study the behavior of the applications under different network conditions. To this end, we run the `tc-netem` tool on the Linux gateway to progressively decrease the available bandwidth from 100 to 5 Mbit/s, impose additional latency from 10 to 300 ms, or 1–10% packet loss. For Stadia and GeForce Now, we replicate all the experiments using both the PC and the smartphone. Moreover, we also perform all experiments with the Stadia dongle. Finally, we take Stadia as a case study to understand the behaviour with different mobiles networks. To this end, we perform different gaming sessions with the PC and emulated on the Linux gateway different mobile networks using ERRANT [24]. ERRANT is a state-of-the-art network emulator which imposes realistic network conditions based on a large-scale measurement campaign under operational mobile networks. ERRANT can reproduce the variability of conditions intrinsically rooted in mobile networks due to different operators, radio access technologies (RATs) (i.e., 3G or 4G), or signal quality (e.g., bad quality due to weak signal). ERRANT comes with 32 network profiles describing the typical network conditions observed in different European operators under 3G and 4G. We also use the ERRANT speedtest training dataset to study the possibility of using Stadia on different conditions under mobile networks.

**Table 1.** Overview of the measurement campaign.

| Application | PC | Smartphone | Dongle | Quality Levels | Server Location | Traffic Shaping | Mobile Networks | Total Tests |
|---|---|---|---|---|---|---|---|---|
| Stadia | ✓ | ✓ | ✓ | 3 | | ✓ | ✓ | 94 |
| Geforce Now | ✓ | ✓ | | 2 | ✓ | ✓ | | 71 |
| PS Now | ✓ | | | | | ✓ | | 60 |

In total, we collect 225 packet traces, summing to 390 GB of data. We share with the research community a sample of these traces from the three services at [5]. Then, we analyze the traffic traces using the Wireshark packet analyzer. https://www.wireshark.org/ (1 October 2021). We also use Tstat [25], a passive meter, to obtain flow-level logs summarizing the observed TCP/UDP flows. Finally, we use the Chrome debugging console to study Stadia and the disk log files for GeForce Now.

## 4. Results

We now illustrate the findings that we obtain from the analysis of the collected network traces. We first show which network protocols that each service employs for streaming and signalling (e.g., user's commands), and analyze in detail the different approaches used for audio and video transmission. We then provide quantitative figures for the volume of traffic the services generate at different video quality levels and study the impact of mobile network scenarios. Finally, we study the contacted servers in terms of autonomous systems (ASs) and RTT distance, and discuss how the infrastructures are organized.

### 4.1. Employed Protocols

In this section, we describe the protocols used by the three cloud gaming providers to stream multimedia content and transmit control data for, e.g., session setup and users' commands. Table 2 provides an overview of the protocols that we observe, as well as the employed codecs.

*Stadia:* The service from Google uses the most standard protocol mix, as it relies on WebRTC [26]. In a few words, WebRTC is a set of standard application programming interfaces (APIs) that allow real-time communication from browsers and mobile applications. It establishes sessions using the Datagram Transport Layer Security (DTLS) protocol for key exchange. The multimedia connection between the client and server is set up using interactive connectivity establishment (ICE), which in turn relies on the session traversal utilities

for network address translators (STUN) and the traversal using relays around NAT (TURN) protocols for NAT (network address translator) traversal. We find that Stadia uses WebRTC with no substantial modifications, both from the browser and mobile application. The traffic captures using the dedicated dongle device (Chromecast) confirm that the observed traffic is consistently compatible with WebRTC. When the multimedia session begins, the client starts a normal WebRTC peer connection to the server, creating a UDP flow in which DTLS, STUN and RTP are multiplexed according to the RFC 7893 [27]. RTP is used for multimedia streaming, while DTLS carries the user's input. We also observe packets of the real-time control protocol (RTCP) [3,28], used to exchange out-of-band statistics between the sender and the receiver of a multimedia stream. The RTCP payload is encrypted to enhance users' privacy, preventing the in-network devices from using it for quality of service monitoring.

*GeForce Now:* It adopts a different approach. The server is first contacted using the TLS (over TCP) protocol to set up the session. Interestingly, the Client Hello messages contain the Server Name Indication extension, which allows us to infer the server hostname (see Section 4.6 for details). Then, the client opens multiple UDP channels directly, without relying on the standard session establishment protocols (ICE, STUN, and TURN). Only the first packet from the client contains an undocumented hello message. Each inbound flow then carries a standard RTP stream. The client sends the user commands on a dedicated UDP flow using an undocumented protocol. All flows use fixed ports on the client-side, in the range 49,003–49,006, while they vary on the server-side. Here, we do not observe the presence of the RTCP protocol.

*PS Now:* This service adopts a completely custom approach, with no standard in-clear header. The client opens a UDP connection towards the server without relying on any documented protocol, and, as such, we can only analyze the raw payload of packets. Still, complex manual work allowed us to catch at least the high-level encapsulation schema that we briefly describe here. The first byte of the packet is used to multiplex multiple data channels. The channel 0 is used for signalling and user's commands, while 2 and 3 are used for multimedia streaming from the server. This is confirmed by the plausible packet size and inter-arrival distributions, and allows us to infer which kind of multimedia content is carried on each channel, as we illustrate in Section 4.3.

*Take away: Stadia relies on WebRTC for streaming and session setup. GeForce Now uses RTP with no standard session establishment protocol. PS Now employs a fully-custom approach.*

**Table 2.** Protocol usage for different gaming session components.

|  | Stadia | GeForce Now | PS Now |
| --- | --- | --- | --- |
| Streaming | RTP (and RTCP) | RTP | Custom (UDP) |
| Player's input | DTLS | Custom (UDP) | Custom (UDP) |
| Session setup | DTLS, STUN | TLS | Custom (UDP) |
| Network Testing | RTP | Iperf-like | Custom (UDP) |
| Video Codec | H.264, VP9 | H.264 | - |

*4.2. Network Testing*

All three services have built-in functionalities to probe the network between the client, and (multiple) gaming server machines to determine if the conditions are sufficient for a stable gaming session. In a few words, the client applications perform a speed test-like measurement to estimate the network delay and bandwidth. We notice that the network testing is not performed consistently on each session startup, but the applications tend to re-probe the network only after a variation of the client IP address.

Stadia performs a speed test based on RTP packets carried over a session established using the standard WebRTC APIs for the multimedia streams. The server (not necessarily the same used for the subsequent gaming session) sends 5–6 MB of data to the client, resulting in a UDP session which is 5–60 s long, depending on the network conditions. The

RTP packets are large, around 1200 bytes on average, but we cannot inspect their payload, since it is encrypted.

GeForce Now uses a schema similar to the one used in the popular tool Iperf (https://iperf.fr/, accessed on 1 October 2021),in which the client sets up a network test over a UDP channel on the server port 5001 (the same port used by Iperf). The first few packets carry JSON-encoded structures to set up the test. In case the test includes a latency measure, the last flow packets indicate the measured RTT samples. In case the test is only for bandwidth, we observe a stream of large-sized UDP packets lasting 5–10 s. Again, the testing server is different from the one used for the subsequent gaming session. The inspection of the JSON messages allows us to understand that the client probes the latency towards multiple alternative measurement servers.

PS Now adopts a fully custom approach again. At the beginning of each gaming session, the client performs a few-seconds long bandwidth test using a custom or fully encrypted protocol running over UDP. We cannot infer any information from the packets, for which we only observe that they all have size 1468 bytes. The test is performed towards a server different from the one used for the proper gaming streaming session. We note similar additional streams consisting of few packets toward a handful of other servers that we conjecture are used to probe the latency towards more endpoints.

**Privacy concerns:** While analyzing the GeForce Now network testing mechanism, we notice that the client-side control packets used to set up the test expose the user to a severe privacy concern [29]. The user ID is sent in clear into the UDP packet, allowing an eavesdropper to uniquely identify a user, even if they change their IP address or are roaming on another network. We compared the user ID to the user account number that we obtained on the NVIDIA website profile management page, and they match, confirming that the identifier is uniquely associated with the account. Following the best practices for these cases, we signalled the issue to NVIDIA before making our paper public, which plans to resolve it using one of the following updates.

*Take away: Stadia uses an RTP stream for testing the network, while GeForce Now relies on an Iperf-like mechanism. Again, PS Now employs a fully custom approach. We found a severe privacy leakage in the current GeForce Now implementation that allows an eavesdropper to obtain the user identifier.*

*4.3. Multimedia Streaming*

We now analyze how the three services stream the multimedia content (audio and video) from the gaming server to the client. In the case of Stadia and GeForce Now, we will provide figures extrapolated inspecting the RTP headers, while for PS Now, we can separate the different streams by looking at the first byte of the UDP payload, as mentioned in the previous section. In the last row of Table 2, we report the employed video codecs as we extract from the browser/application log files. The widespread H.264 codec is used by both GeForce Now and Stadia, employing the newer VP9 if the device supports it. For PS Now, we could not obtain any information about the codecs.

Stadia relies on the WebRTC APIs, and, as such, the multimedia streaming follows its general principles. A single UDP flow carries multiple RTP streams identified by different source stream identifiers (SSRC). A stream is dedicated to the video, while another one to the audio track. We also find a third flow used for video retransmission, as we confirm using the Chrome debug console (the associated RTP stream is found to have mimeType video/rtx). During most of the session, it is almost inactive. At certain moments, the flow becomes suddenly active, carrying large packets containing video content. Moreover, this behaviour co-occurs with packet losses and bitrate adaptations on the video stream, as we expect for a video retransmission feature.

GeForce Now again relies on RTP for multimedia streaming, as described in the previous section. Differently from Stadia, it uses separate UDP flows for the different multimedia tracks, whose client-side port numbers can be used to distinguish the content as NVIDIA publicly declares (https://nvidia.custhelp.com/app/answers/detail/a_id/4504/~/how-

can-i-reduce-lag-or-improve-streaming-quality-when-using-geforce-now, accessed on 1 October 2021). On port 49, 005, a UDP flow carries a single RTP stream for the inbound video. The audio is contained in a UDP flow on port 49, 003, in which we find two RTP streams active at the same time.

Regarding PS Now, we cannot find any header belonging to publicly documented protocols. However, the inspection of several packet captures allows us to infer the encapsulation schema used by the application. A single UDP flow carries all multimedia streams. To multiplex the streams, the first byte of the UDP payload indicates the channel number, followed by a 16-bit long sequence number. Channel 2 carries the video stream, as we can conclude by looking at packets' packet size and inter-arrival time. Channel 3 carries the audio track as the packets are small and fixed-sized (250 B), and arrive at a constant pace of one every 20 ms. We also find channel 0, especially at the beginning of the flow, which we conjecture is used for signalling. Finally, channel 12 seldom becomes active, especially in the correspondence of large packet losses. As such, we conjecture that it is used for video retransmission or some form of forwarding error correction (FEC), similarly to the Stadia approach.

*Take away: For Stadia, a single UDP flow carries separate streams for audio, video and video retransmissions. GeForce Now uses multiple UDP flows on different client-side ports. In PS Now, a single UDP flow appears to carry an audio, a video and a retransmission/FEC stream.*

### 4.4. Network Workload

We now focus on the workload imposed on the network by users playing on cloud gaming services. We start our analysis with Figure 2, in which we show the evolution of a gaming session of around 10 min for each service. We made the corresponding packet traces available to the community at [5]. The picture reports the bitrate of the inbound traffic, due almost exclusively to the video multimedia stream. We first notice that Stadia (Figure 2a) has a constant bitrate, while for GeForce Now and PS Now (Figure 2b,c respectively), it is considerably more variable. Especially for GeForce Now, we can root this in the different video codecs, as we describe later in this section. Indeed, the application logs confirm that the variations in the bitrate are not caused by resolution adjustments or codec substitution. Looking at Stadia, the role of the video retransmission stream (green dashed line) is clear, which becomes active in the correspondence of impairments in the main video stream (solid red line). We notice a very similar behaviour in PS Now, which allows us to conjecture the presence of an analogous mechanism.

We summarize the network workload, showing in Figure 3 the empirical cumulative distribution function of the video bitrate that we observe. For Stadia and GeForce Now, we report separate distributions for different video resolutions thanks to Chrome debug console and application logs, respectively. For PS Now, we only report the overall distribution as we cannot extract any statistics from the client app. As mentioned before, Stadia exhibits a rather constant bitrate (Figure 3a). The service allows three streaming resolutions, namely 720p, 1080p, and 4K (2160p), whose average bitrate figures are 11, 29 and 44 Mbit/s respectively. This is consistent with what is declared in documentation. https://support.google.com/stadia/answer/9607891, accessed on 1 October 2021. Stadia employs both H.264 and VP9 video codecs, with 4K streaming using uniquely VP9.

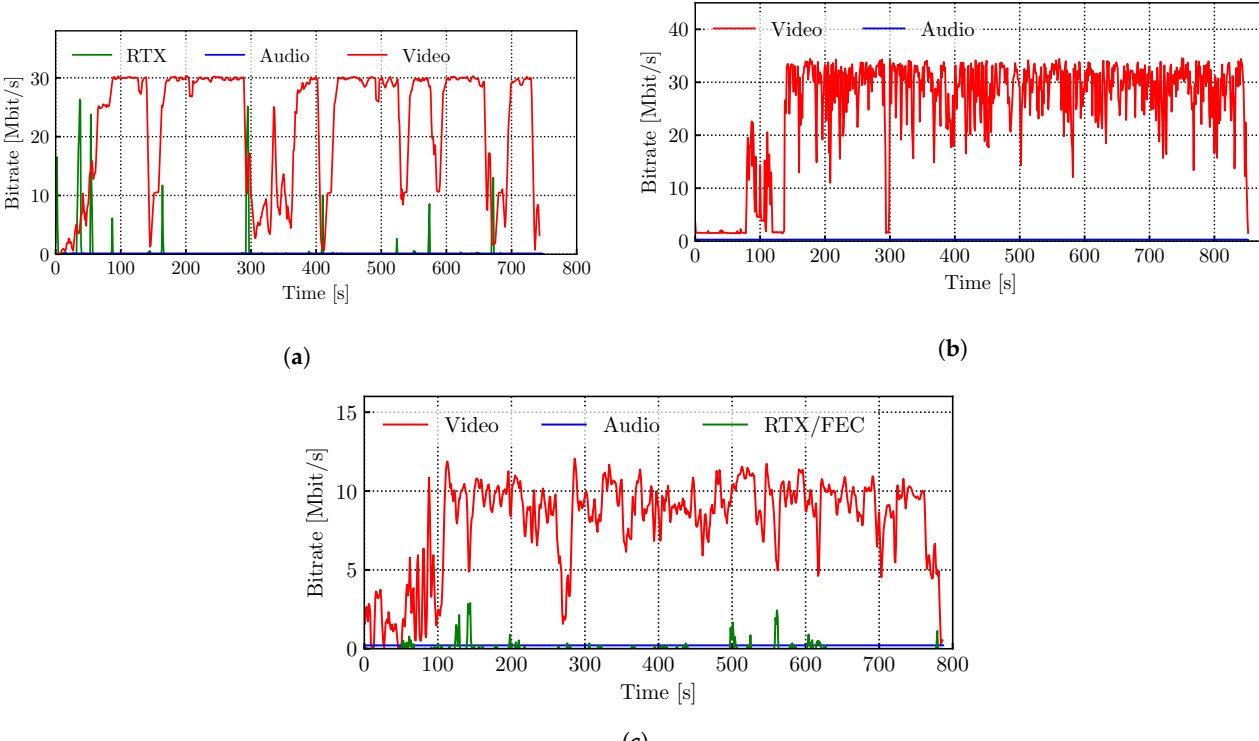

**Figure 2.** Examples of temporal evolution of bitrate for three gaming sessions. (**a**) Stadia (1080p). (**b**) GeForce Now (1080p). (**c**) PS Now.

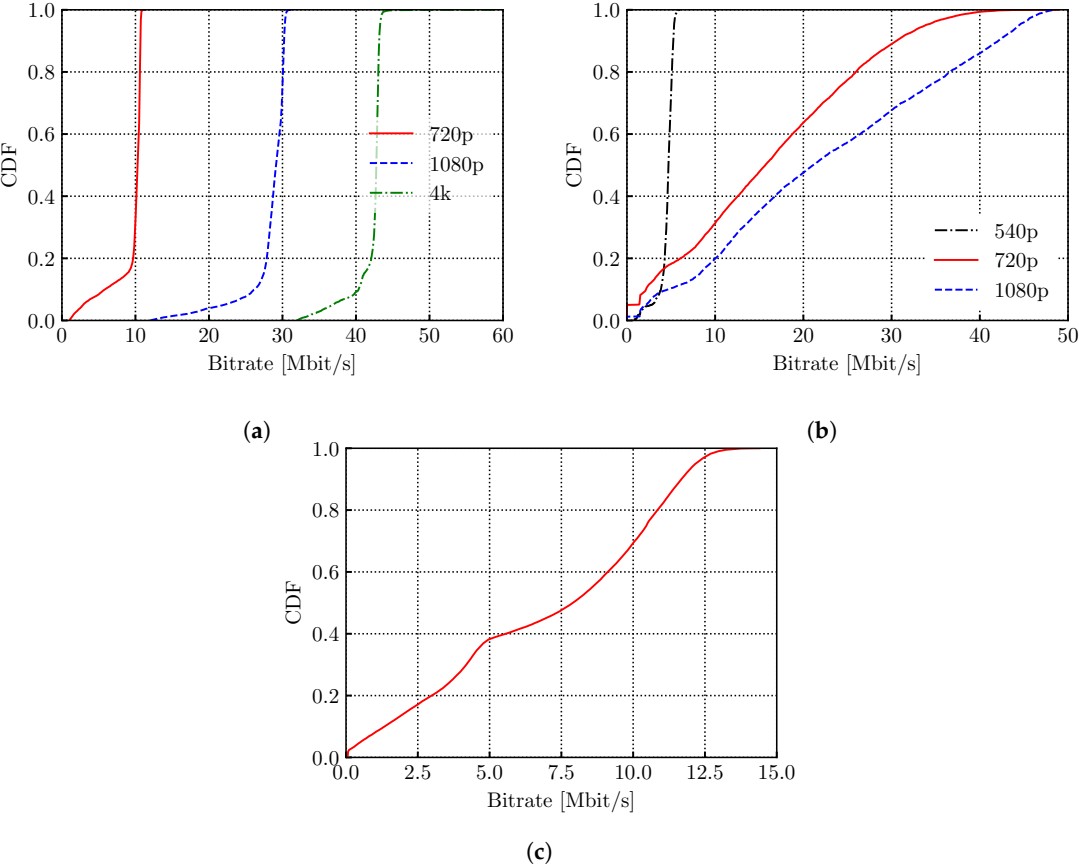

**Figure 3.** Cumulative distribution of the video bitrate, for different quality video levels. (**a**) Stadia. (**b**) GeForce Now. (**c**) PS Now.

Different is the picture for GeForce Now, shown in Figure 3b. The service allows several video resolutions, both in 16:9 and 16:10 aspect ratios. Here, we report the bitrate of the lowest (720p) and the highest (1080p) resolutions available for the 16:9 aspect ratio. Moreover, we show the 540p resolution, which is only adopted automatically in case of bad network conditions, that we trigger imposing a bandwidth limit on the testing machine. The figure shows that the bitrate has large variability, especially for 720p and 1080p. On median, 720p (1080p) consumes 15 (20) Mbit/s, which is consistent with what is declared on the system requirements of the service (https://www.nvidia.com/it-it/geforce-now/system-reqs/, accessed on 1 October 2021). However, the interquartile ranges (IQRs) are in the order of 15 Mbit/s, much more than the 2–3 Mbit/s IQRs observed in Stadia. The bitrate reaches peaks of more than 30 and 40 Mbit/s for 720p and 1080p, respectively. Without access to the unencrypted raw video stream, we conjecture that GeForce Now makes a highly-dynamic use of the H.264 compression parameters to adapt to different video characteristics (static/dynamic scenes) and network conditions. Indeed, our experiments with limited bandwidth show that, for example, GeForce Now can sustain a 1080p video stream also with less than 15 Mbit/s available bandwidth without dropping the frame rate, likely adjusting the H.264 compression parameters.

Finally, Figure 3c shows the bitrate distribution for PS Now. Given the lack of application settings or debug information, we only show the overall distribution of the bitrate. The online documentation recommends a minimum bandwidth of 5 Mbit/s and states that video streaming has a 720p resolution. However, we observe the bitrate reaching up to 13 Mbit/s, with a consistent variability. Interestingly, when we impose a 10 Mbit/s or lower bandwidth limitation on the network, the bitrate adapts consequently (see the peak in the distribution at 5 Mbit/s). However, we cannot link it with a resolution lower than 720p.

We now study the impact of packet losses on cloud gaming, focusing on Stadia as a case study. As described in Section 3, we run experiments in which we enforce artificial packet loss via the `tc-netem` tool, ranging from 1% to 10%. We then study how the application reacts to these scenarios, focusing on the sending video bitrate and quality level. In Figure 4, we show the distribution of bitrate for experiments with 1, 5 and 10% packet loss, and report the bitrate of sessions with no packet loss as a reference (solid red line). We notice that a packet loss of 1 and 5% (blue and green dashed lines, respectively) does not cause significant bitrate variations. Indeed, the bitrate reaches 30 Mbit/s, and we notice that the game is running at 1080p most of the time, with some 720p periods. We notice a constant activity on the retransmission stream, used to recover lost packets. Different is the case with 10% packet loss (yellow dashed line). In this case, Stadia keeps the video resolution at 720p most of the time, and, as such, the bitrate is limited to 10 Mbit/s. We notice a high activity again on the retransmission stream, which, however, is not deemed sufficient to go for higher video resolutions. In a few cases, 1080p is achieved, but for short 5–10 s periods. In all the cases, the games were still playable and enjoyable, as reported by the volunteers.

*Take away:* Stadia *has three video resolutions, with a rather constant bitrate of (median of 11, 29 and 44 Mbit/s). Packet losses up to 5% do not cause a reduction in resolution.* GeForce Now *streams video at many different resolutions, each having a large variability in the bitrate, reaching up to 45 Mbit/s.* PS Now *does not exceed 13 Mbit/s, and we cannot infer the employed quality levels.*

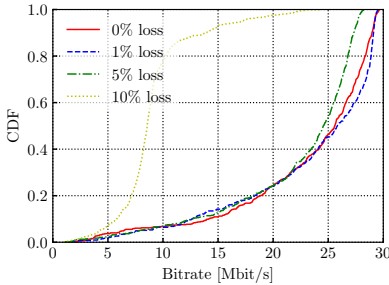

**Figure 4.** Bitrate distribution with different artificial packet loss.

### 4.5. Cloud Gaming under Mobile Networks

We now investigate the case of mobile networks to understand to what extent cloud gaming is feasible and what is the reached quality level. Moreover, we are interested in understanding the impact of variable network conditions, and how applications react to network impairments. Here, we focus on Stadia, which streams video content at fixed bitrates, requiring strict bandwidth constraints.

We first build on a large-scale measurement campaign run on our previous work [24], to study to what extent current mobile networks are ready to sustain the load and offer suitable conditions for cloud gaming. The dataset that we use includes more than 100k speed test measurements from 20 nodes/locations equipped with SIM cards of 4 operators, located in 2 distinct countries (Norway and Sweden). Each experiment measured latency, downlink, and uplink capacity using multiple TCP connections towards a testing server located in the same country. Moreover, the dataset indicates physical-layer properties such as radio access technology (3G or 4G) and signal strength. The measurement campaign spans the whole of 2018 and includes experiments on different hours of the day and days of the week. We use the dataset to understand how a Stadia cloud gaming session would have performed with the measured network characteristics. This is possible thanks to the steady network usage of Stadia, where different video resolutions utilize almost fixed bandwidth. As such, given the network conditions measured on the speedtest measurements, we consider running Stadia *feasible* if there is at least 10 Mbit/s available bandwidth. In the case where available bandwidth is in [10–30) Mbit/s, we conclude that a user could only reach a video resolution of 720p. When bandwidth is in the range [30–44) Mbit/s, a better 1080p video could be transmitted, while more than 44 Mbit/s allow the user to receive a 4*K* quality video properly.

Figure 5 shows the reached video quality levels, offering a breakdown on 3G and 4G and different signal qualities as measured by the node's radio equipment (The measured Received signal strength indication (RSSI) is mapped to quality levels, as recommended at https://wiki.teltonika.lt/view/Mobile_Signal_Strength_Recommendations accessed on 1 October 2021). The figure shows how a poor 3G link does not allow the use of Stadia in most cases, as the available bandwidth is below 10 Mbit/s. The picture changes with medium signal quality, where using Stadia is possible most of the time, with the lowest 720p resolution. An excellent 3G connection allows higher resolution in less than 10% of cases. With 4G, the network generally offers higher bandwidth, and 1080p and 4K are possible, especially with good signal quality. Indeed, 4G links with high signal strength can sustain Stadia 4K streaming 40% of the cases, and only 38% of the times, is the client limited to 720p.

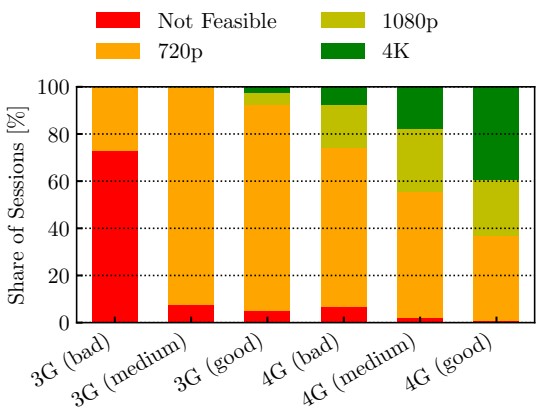

**Figure 5.** Estimated resolution of Stadia sessions on mobile networks.

Next, we study the capability of the gaming platforms to offer a good gaming session and cope with different mobile network conditions. To this end, we use the PC to run

a gaming session and track different quality metrics about the frame rate, expressed in frames per second (*FPS*), the packet loss, and the video resolution quality experienced during the gaming sessions. We focus on Stadia as a case study, since it is available on many devices.

To study the gaming session in a controlled mobile network environment, we emulate different mobile network conditions on the Linux gateway by using ERRANT. In detail, we perform experiments by using a 4G good network profile as, currently, it is the best quality and most widespread network, and with a 3G good network profile, as it is the lowest quality profile showing enough bandwidth to support a gaming session. To reproduce the mobile network variability, we set ERRANT to resample the network condition every 10 s; i.e., we pick and apply new constraints for the download rate, upload rate, and latency from the selected profile.

Figure 6 reports an example of the frame rate (right y-axis), the packet loss (left y-axis), and the video resolution quality when a sudden change happened during a gaming session while using the 4G good profile. Interestingly, we experienced stable performance for most gaming sessions, with no packet loss, 60 FPS, and 1080p resolution. Only when ERRANT picks a download bandwidth below 10 Mbit/s do we experience, for a short time, a reduction in frame rate and resolution down to 20 FPS and 720p, respectively, with there seldom being an increase in packet loss. Interestingly, the Stadia platform can quickly react and adapt itself by reaching a frame rate of 60 FPS and resolution at 720p, rising again to 1080p as soon as a better bandwidth is available. With a 3G good network profile, instead, we could not run an entire gaming session, as the network variability introduced by the mobile network caused the game to stop suddenly. This shows how the promising benefits of these solutions currently have limitations that could be overcome with the increasing popularity of fast mobile network technologies—4G and, in particular, 5G.

*Take away: Stadia cannot run in most of the cases with a bad 3G network. With proper signal strength, 3G allows a resolution of 720p. With 4G, it is possible to achieve higher quality levels, and good signal strength allows 4K video resolution in 40% of situations. Controlled experiments reveal how Stadia quickly reacts to deteriorated network conditions.*

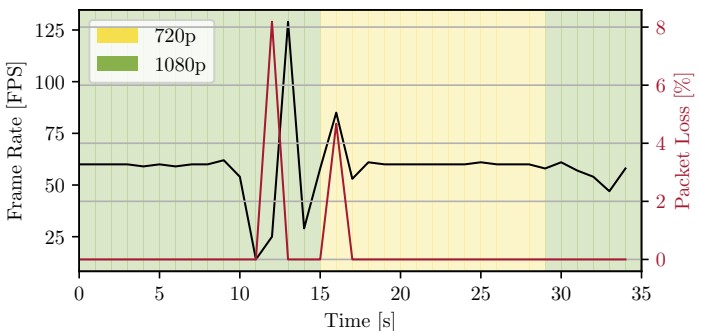

**Figure 6.** Example of quality-level reduction in Stadia, in terms of frame rate and video resolution subsequent to packet losses.

### 4.6. Location of Gaming Machines

This section provides a preliminary investigation of the cloud gaming infrastructure regarding the number and location of game servers and employed domains, as observed from our location. This can be useful to identify cloud gaming traffic for, e.g., traffic engineering or content filtering.

We first focus on the remote gaming machines, analyzing when the server IP addresses the client applications contact, summarized in Table 3. Indeed, after the initial setup phase, the client exchanges traffic (almost) uniquely with a single server where the gaming is likely executed (we cannot infer if the server IP address acts as an ingress load balancer or reverse proxy). Considering Stadia, at each session, we contacted a different server—i.e.,

we performed 74 sessions and reached 74 different server IPs. They lay on 22 subnets/24, all belonging to the Google 15129 AS (we map an IP address to the corresponding AS using an updated RIB from http://www.routeviews.org/ accessed on 13 January 2021). Different is the case for GeForce Now and PS Now, for which in roughly 50% of the cases, we contacted a server IP we had already observed. GeForce Now servers lay on 23 subnets belonging to three different ASes, all controlled by NVIDIA. Recall that we used all the 14 available NVIDIA data centres in our experimental campaign by instrumenting the client application. Finally, all 36 server IPs for PS Now belong to only 2 subnets/24 from the 33353 Sony AS. In terms of latency, additional `ping` measures show that Stadia and PS Now servers are 6–8 ms away from our location, with Central European GeForce Now in the order of 15–20 ms. However, we cannot link this to the service quality or infrastructure deployment, since we perform measurements from a single vantage point. We did not qualitatively observe a significant lag between users' commands and game response without traffic shaping.

**Table 3.** Gaming servers characterization.

|  | **Servers** | **Subnets** | **ASNs** | **Owner** |
|---|:---:|:---:|:---:|:---:|
| Stadia | 74 | 22 | 15169 | Google |
| GeForce Now | 37 | 23 | 11414, 20347, 50889 | NVIDIA |
| PS Now | 36 | 2 | 33353 | Sony |

Finally, we analyze the domains that the applications contact during the gaming sessions. We extract them by inspecting the client's domain name system (DNS) queries, before opening a new connection and extracting the Server Name Indication (SNI) field from the Transport Layer Security Security (TLS) Client Hello messages. In the case of Stadia, we only find `stadia.google.com` as domain-specific to this service. Indeed, the client application contacts a dozen of other domains, but those are shared with all Google services (e.g., `gstatic.com` and `googleapis.com`), and, as such, they are not specific of Stadia. Regarding GeForce Now, the application contacts the general `nvidia.com` domain as well as the more specific `nvidiagrid.net`. Finally, PS Now relies on the `playstation.com` and `.net` domains and their sub-domains, which are, thus, not specific to the PS Now service. Interestingly, GeForce Now is the only service that uses domains to identify gaming machines, using subdomains of `cloudmatchbeta.nvidiagrid.net`. Indeed, for the other services, the domains that we find are associated uniquely with the control servers—used for login, static resources, etc.—while gaming machines are contacted without a prior DNS resolution.

*Take away: We seldom observe repeated IP addresses of the gaming machines for the three services. They are all located in the AS of the respective corporate. GeForce Now is the only one that identifies gaming machines via DNS domain names.*

## 5. Conclusions and Future Directions

In this paper, we showed important aspects of the recently launched services for cloud gaming, namely Stadia, GeForce Now, and PS Now. Millions of users could potentially use them shortly, accounting for a large part of Internet traffic. Indeed, Stadia and GeForce Now stream data up to 44 Mbit/s, which is, for instance, much higher than a 4K Netflix movie. However, fast 4G mobile networks can often sustain this load, while traditional 3G connections struggle. Cloud gaming applications rely on the standard RTP protocol for multimedia streaming, except for PS Now, which does not use any documented protocol.

The services mentioned above entered the market between 2019 and 2020, and, as such, a lot of work is still to be done for characterizing them and understanding their impact on the network. Future directions include studying their infrastructures from different points of view worldwide to find similarities and differences in the deployments. Moreover, we

believe that it is crucial to conduct campaigns aiming to measure the subjective quality of experience (QoE) enjoyed by the users with human-in-the-loop controlled experiments, and with AI-bot players [30]. Finally, Microsoft and Amazon will release their cloud gaming platforms, namely xCloud, and Luna, which are in early deployment at the time of writing this article. Any future work on this topic must include them.

**Author Contributions:** Conceptualization, M.T., L.V. and D.G.; Investigation, A.D.D., G.P. and M.T.; Methodology, M.T. and A.D.D.; Software, A.D.D.; Supervision, M.T., L.V. and D.G.; Writing—Original Draft, A.D.D., G.P. and D.G.; Writing—Review and Editing, M.T., L.V. All authors have read and agreed to the published version of the manuscript.

**Funding:** This research received no external funding.

**Data Availability Statement:** The supplementary material containing sample traces regarding the network traffic generated while playing with GeForce Now, PS Now, and Stadia are available at [5].

**Acknowledgments:** The research leading to these results is supported by the SmartData@PoliTO center for data analysis and Big Data technologies.

**Conflicts of Interest:** The authors declare no conflict of interest.

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
