# Peer review of "A Network Analysis on Cloud Gaming: Stadia, GeForce Now and PSNow"

_2673-8732, doi:10.3390/network1030015_

Round 1

Reviewer 1 Report

The paper is very well organized, technically sound, addresses a very recent topic and the English is very good.
Is RTCP used for QoS monitoring?  It could be used for triggering codec changes.  It would also be interesting to check if there are RTCP feedback messages [RFC 4585, RFC 5104] for triggering retransmissions.  If RTCP is not used, you should refer it in the paper, as RTCP is a companion protocol to RTP. You might add to Table 1 the signaling protocol and information about flows being encrypted or not.

Several additional minor corrections are suggested:
expand acronyms on first use: HTTP, P2P, GPU, RTP, RAT, DTLS, ICE, STUN, TURN, NAT, DNS, SNI, TLS
line 15: "in bandwidth-hungry" is an adjective that is missing the corresponding noun. I suggest replacing with "in bandwidth-hungry flows"
line 74: "Section 2" -> "Section 4"
line 259-260: "to publicly documented protocol" -> "to a publicly documented protocol" or "to publicly documented protocols".
line 278: footnote 3 is incorrect.
line 307-308: "observed Stadia" -> "observed in Stadia".
line 325: "ranging from 1 to 10" -> "ranging from 1% to 10%"?
line 369: "node radio equipment" -> "node's radio equipment"
figure 5: It is not clear what "%" in the vertical axis refers to. I suggest labelling with something like: Fraction of sessions [%].
line 391: "right x-axis" -> "left y-axis in black"
line 391-392: "left x-axis" -> "right y-axis in red"
line 417: "The lay" -> "They lay"
References [4] and [5] are duplicates.

Author Response

We would like to thank the reviewer for her/his appreciation for our work and the provided comments and suggestions. Please find the point-to-point responses to each reviewer concern below.  

  • We followed the suggestion of the reviewer regarding the RTCP protocol. We do observe it for Google Stadia, even if the payload is encrypted. In the revised version of the manuscript, we explicitly state it (in the text and table 1) and observe that it cannot be used at the network level for passive monitoring given the obfuscated payload.
  • We now expand all the acronyms at the first use and correct all the suggested typo. We further proofread the paper to fix additional typos and unclear sentences.

Reviewer 2 Report

This paper analyzes communication conducted by major cloud gaming platforms by collecting packet traces in various conditions.
The reported results are interesting and will be helpful for future network design and management. 

Overall the paper is well written.  Take away at the end of each subsection in section 4 is very useful for readers to understand the essential findings in this research.

One minor comment:
In line 74,   "while Section 2 illustrates" should be "while Section 4 illustrates."

Author Response

We would like to thank the reviewer for her/his appreciation for our work and the provided comments and suggestions. Please find the point-to-point responses to each reviewer concern below.  

  • We further proofread the paper to fix additional typos and unclear sentences.